# Variability in head computed tomography use for minor head injury after ground-level falls in the emergency department: A subanalysis of EPI-TC study

Xavier Dubucs[1,2,3]*, Frederic Balen[1,2], Pierre-Hugues Cormicael[4], Axel Benhamed[3], Valérie Boucher[3], Éric Mercier[3,5], Sandrine Charpentier[1,2], Marcel Émond[3,5], IRU EPI-TC group¶

1 Pôle médecine d'urgence, Centre Hospitalo-Universitaire Toulouse, Toulouse, France, 2 Centre d'Épidémiologie et de Recherche en santé des Populations, INSERM - UMR 1295, Université de Toulouse, Toulouse, France, 3 Centre de recherche du Centre Hospitalo-Universitaire de Québec-Université Laval, Québec City, Canada, 4 Centre d'excellence sur le vieillissement de Québec, Québec City, Canada, 5 VITAM – Centre de recherche en santé durable, Québec City, Canada

¶ Membership of the IRU EPI-TC group is listed in the Acknowledgments.
* xavier.dubucs@gmail.com

## Abstract

### Objective

The aim of this study was to assess the variation in the use of head computed tomography (CT) scan in patients attending EDs with ground-level fall-related minor head injury. Secondary objectives were: i) to measure the prescription rate of appropriate head CT scan, ii) to identify patients' and EDs characteristics associated with head CT scan prescription iii) to explore potential correlation between head CT scan use and traumatic intracranial hemorrhage (ICH) yield rate in this population.

### Materials and methods

This research was a planned sub-analysis of a cross-sectional prospective multi-centric study performed in 63 EDs in France (EPI-TC study). Patients sustaining ground-level fall-related with minor head injury were included in this sub-analysis. The main outcome was head CT scan performed during ED stay. Variations in the use of head CT scan were described depending on each ED and French region. Multiple fixed effects mixed logistic regression model was performed to identify factors associated with head CT scan.

### Results

A total of 631 patients admitted for head injury after ground-level fall were included. Median age was 79 [63–88] years. A head CT scan was performed in 409 patients (64.8%, CI95% 61.0–68.5); 19.6% (CI95% 15.8–23.7) of them were appropriated;

 

**Data availability statement:** All relevant data are within the manuscript and its Supporting information files.

**Funding:** This study was funded by the Société Française de Médecine d'Urgence (French Society of Emergency Medicine) through the URI-2023 Research Network (Initiatives de Recherche aux Urgences 2023 to X.D). There was no additional external funding received for this study. All the funding or sources of support received during this study had no role in the design, data collection, data analysis, and reporting of this study.

**Competing interests:** The authors have declared that no competing interests exist.

and 29 (7.1%, CI95% 4.8–10.0) showed a traumatic ICH. At regional level, head CT scan prescription rate ranged from 45.5% (CI95%: 24.4–67.8) to 84.6% (CI95% 54.5–98.1). Head CT scan use was not correlated with the yield rate of traumatic ICH. In multivariable analysis, preinjury antiplatelets (OR 29.2, CI95%: 12.2–69.9), anticoagulants (OR 69.9, CI95%: 20.0–243.9), syncope (OR 6.9, CI95%: 2.0–24.2), post-trauma amnesia (3.2, CI95%: 1.0–10.5) and post-trauma loss of consciousness (OR 5.6, CI95%: 2.0–15.9) were associated with head CT scan use.

## Conclusions

Head CT scan use in patient presenting to EDs with head injuries after ground-level falls is highly variable. High rate of head CT scan use is not correlated with high traumatic intracranial hemorrhage yield rate. The use of a clinical decision rule dedicated to this population would be suitable for harmonizing our practices.

## Introduction

Traumatic Brain Injury (TBI) is a common cause of admission to Emergency Departments (ED) and most of them are classified as mild TBI [1]. Non-contrast head computed tomography (CT) scan is the gold standard in TBI to detect traumatic Intracranial Hemorrhage (ICH). However, recent studies have shown an overuse of head CT scan up to 15%in case of mild TBI [2]. This is particularly significant given that repeated exposure to x-ray source radiation may be associated with an increased risk of cataracts and neoplasia [3,4]. It also represents a substantial financial burden for healthcare systems and may increase patients' length of stay. A recent survey has reported a wide variation in the use of head CT scan in patients with mild TBI across many European countries [5]. Several environmental factors may be related to this variation such as the existence of national guidelines, their adherence as well as patients' characteristics [5,6]. This is of the upmost importance to homogenize practices given that ground-level falls became the leading cause of head injury, especially in older patients [7]. Indeed, it has been shown that 40% of head CT scans performed in patients presenting minor head injury were related to ground-level fall [8]. Furthermore, recent literature has shown an increased number of older patients with TBI admitted to EDs. This trend is associated with a significant increase of CT scan in this population [9,10]. However, the assessment of recent clinical practices regarding head CT scan have only been evaluated using surveys and thus data from usual care setting are lacking [5,11]. In addition, the variation of head CT scan in patients with head injury only related to ground-level fall has not been carried out yet.

The aim of this study was to assess the variation in the use of head CT scan in patients attending EDs with ground-level fall-related minor head injury. Secondary objectives were: i) to measure the prescription rate of appropriate head CT scan, ii) to identify patients' and EDs characteristics associated with head CT scan prescription iii) to explore potential correlation between head CT scan use and traumatic ICH yield rate in this population.

## Materials and methods

### Study design and settings

This was a planned sub-analysis of a prospective observational cohort study that collected patients' data over a three-day period (from 06/03/2023 8:00 am to 09/03/2023 8:00 am, continuously H24) in March 2023 across 71 French EDs [12]. The study cohort comprised EDs across France that accepted an invitation from the *Initiative Recherche Urgence* (IRU; Emergency Research Initiative) network of the *Société Française de Médecine d'Urgence* (SFMU; French Society of Emergency Medicine) to participate in this study. The IRU is a research group that belongs to the French National Society of Emergency Medicine which includes more than 100 EDs in France. In this study, 12 of the 13 regions of Metropolitan France were represented (Auvergne-Rhône Alpes, Bourgogne-Franche-Comté, Bretagne, Centre – Val de Loire, Grand Est, Hauts-de-France, Normandie, Nouvelle Aquitaine, Occitanie, Pays de la Loire, Provence-Alpes-Côte-d'Azur, Île-de-France) and 3 of the 5 French departments overseas (La Réunion, Guadeloupe, Martinique).

### Participants

In this sub-analysis, only patients sustaining a ground-level fall-related with minor head injury (defined as Glasgow Coma Scale (GCS) score ≥13 upon arrival) were included. Patients with unknown head injury kinetic were excluded.

### Data collection and study variables

Upon admission to the ED, the physicians onsite gathered standardized data using the DoqBoard.com observational research platform. This included sociodemographic information, pre-injury use of antiplatelet and anticoagulant medications, fall precipitating factors, symptoms experienced after head injury, clinical examination at the ED and post-emergency outcome (discharge, hospital admission and in-hospital death). Urgent neurosurgical interventions occurring within 48 hours after admission were also reported.

### Outcome measures

The main outcome was non-contrast head CT scan performed during ED stay. The secondary outcome was appropriate head CT scan defined by the French National Guidelines published in 2022. Hence, head CT scan prescription was classified as appropriate if patients undergoing head CT scan showed intermediate or high risk of traumatic ICH [13]. Head CT scan prescription in mild TBI patients across French EDs is determined by the National Guidelines published in 2012 and updated in 2022 [13,14]. These guidelines recommend a head CT scan for mild TBI patients at intermediate risk (patients aged ≥65 years old with antiplatelet therapy, Glasgow Coma Scale (GCS) score <15 two hours after the trauma with a suspected intoxication, that sustained high-energy trauma, or with amnesia ≥30 minutes after the trauma) within 8 hours after being admitted to the ED. In patients presenting a high risk of traumatic ICH (hemostasis disorders, suspected basilar or cranial skull fracture, GCS < 15 two hours after the trauma without intoxication, > 1 vomiting episode, post-traumatic seizures, focal neurological deficit) a head CT scan is required within the first hour.

### Statistical analysis

Categorical variables were reported with their frequency, proportion (%) and 95% Confidence Interval (CI 95%), where appropriate. The description of quantitative variables was recorded through their median and interquartile ranges (IQ1–IQ3) and their mean and standard deviation (SD). Variations in the use of head CT head were described at the ED and at the regional level. In France, regional colleges of emergency physicians that offer annual training conferences and continuing education courses. These courses are not standardized at the national level. National guidelines must be followed by all emergency physicians, but there may be some variability in practices related to the regional training courses offered. La Réunion, Guadeloupe and Martinique were pooled as a unique overseas group of regions to reach a sufficient

number of subjects. The prevalence of traumatic ICH was calculated among patients who underwent a head CT scan. The proportion of appropriate use of head CT scan was calculated based on the number of head CT scan prescriptions. The correlation between the prevalence of traumatic ICH yield and head CT scan prescription rates -both overall and appropriate use- was assessed using Spearman's rank correlation coefficient. Bivariate analyses were performed to compare patients and ED characteristics to the use of head CT scan. These characteristics associated with the use of head CT scans with p-value < 0.2 were then included in a multiple fixed effects mixed logistic regression model. The dependent variable was the use of head CT scan at the patient's level. Random intercepts were modeled per region and per ED sites within regions to allow varying baseline probability to use CT scans. In a first model building step, the association between the use of head CT scans and each patient-level and ED-level characteristics was modeled using mixed logistic regression with the mentioned random effects. A backward stepwise selection method was applied based on the area under the receiver operating characteristic curve (AUC). Missing data rates were reported for each variable and no data imputation was performed. All analyses were performed with Stata 17.0 (StataCorp, Texas, USA). Our results were reported using the STROBE statement.

## Ethics statement

According to French and European law on ethics, all included patients were informed that their anonymized data will be used for the study. The oral non-opposition of all patients was collected, and a written information notice was given to each included patient explaining the objectives of the study and the legislation explaining this research. This non-opposition was documented in the patient's medical file. According to the French ethics and regulatory law (public health code) prospective studies based on the exploitation of usual care data should not be submit at an ethics committee (IRB), but they have to be declared or covered by reference methodology of the French National Commission for Informatics and Liberties (CNIL). Toulouse University Hospital signed a commitment of compliance to the reference methodology MR-004 of the French National Commission for Informatics and Liberties. After evaluation and validation by the data protection officer and according to the General Data Protection Regulation (Regulation EU 2016/679 of the European Parliament and of the Council of 27 April 2016), this study completing all the criteria, it is register in the register of data study of the Toulouse University Hospital (number's register: RnIPH 2022–78) and cover by the MR-004 (CNIL number: 2206723 v 0). This study was approved by Toulouse University Hospital and confirms that ethical requirements were totally respected in the above report.

## Results

A total of 631 patients admitted in one of the 63 participating EDs for head injury after ground-level fall were included (Fig 1). Median age was 79 years (63–88); environmental falls were the leading precipitating factor and more than half of patients (60.5%) had no post TBI symptoms. Baseline patients' characteristics are displayed in Table 1. A head CT scan was performed in 409 patients (64.8%, CI95% 61.0–68.5) and a traumatic ICH was reported in 29 (7.1%, CI95% 4.8–10.0) of them. Among head CT scan prescriptions, 80 of them (19.6%, CI95% 15.8–23.7) were classified as appropriate. No significant correlation was found between overall head CT scan use (Spearman's correlation coefficient: 0.22, p = 0.46) or appropriate head CT scan use (Spearman's correlation coefficient: 0.06, p = 0.86) and traumatic ICH yield rate.

At the regional level, while the main patients' characteristics were similar (Table 2), head CT scan prescription rates were statistically different (p = 0.02) ranged from 45.5% (CI95%: 24.4–67.8) to 84.6% (CI95% 54.5–98.1) (Fig 2). This variation in head CT scan prescription persisted in the subpopulation of patients with no neurological signs, those with no anticoagulants and those with no antiplatelets (Table 3). No statistically significant association was observed between the prevalence of traumatic ICH and regional distribution (p = 0.29) (Fig 2)

In bivariate analysis, ED's characteristics were not associated with head CT scan use (S1 Table). S2 Table displayed bivariate analysis of patients' characteristics regarding the use of head CT scan.

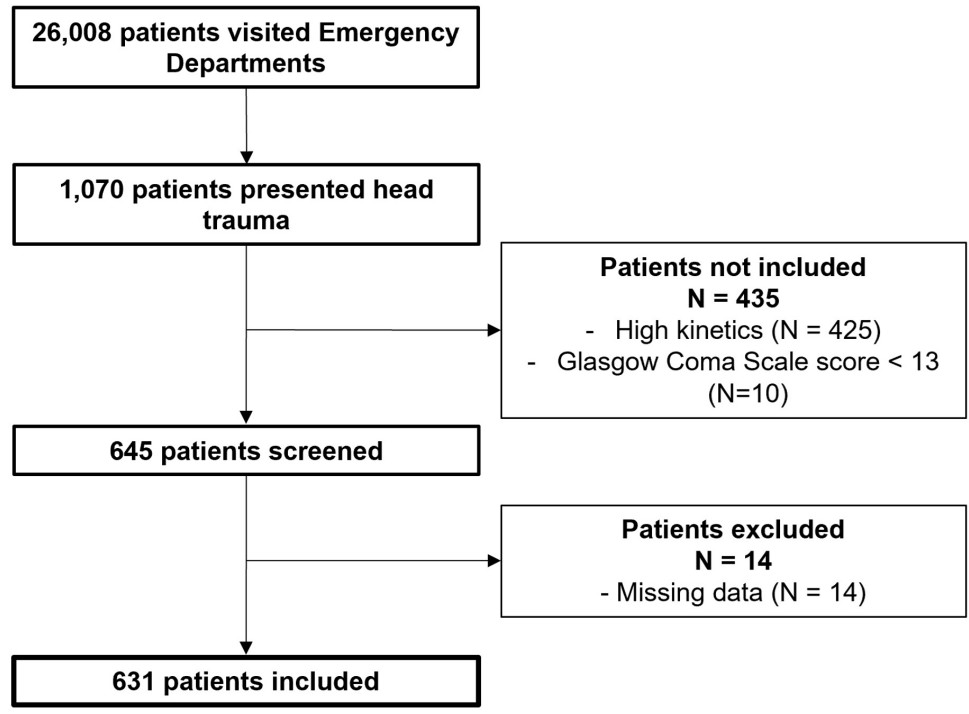

**Fig 1. Flow chart.**

In multivariable analysis, preinjury antiplatelets (OR 29.2, CI95%: 12.2–69.9), anticoagulants (OR 69.9, CI95%: 20.0–243.9), syncope (OR 6.9, CI95%: 2.0–24.2), post-trauma amnesia (3.2, CI95%: 1.0–10.5) and post-trauma loss of consciousness (OR 5.6, CI95%: 2.0–15.9) were associated with the use of head CT scan (Table 4). Multivariable analysis of each subpopulation (patients with no neurological signs, those with no anticoagulants and those with no antiplatelets) are available in S3–S5 Tables (Subgroup multivariate analyses).

## Discussion

These results showed a wide variability in the use of CT scan among patients admitted to the ED after ground-level fall-related minor head injury. Our study suggests that this variation is not explained by patients' characteristics. Several hypotheses may explain these findings. First of all, we assume that national guidelines on the use of head CT scan for patients admitted to ED with mild TBI are not always applied [13]. Assessing the implementation of clinical decision rules in clinical practice remains challenging [15]. For instance, in case of mild TBI, a recent study has shown that the implementation of the Canadian CT Head Rule was associated with a modest decrease in the use of head CT scan (absolute 5.3% reduction in the use of head CT scan) [16]. This decrease widely varied from one ED to another, ranging from +0.4% to −9%. While the performance of the Canadian CT Head Rule has been demonstrated, these results showed the difficulty of applying clinical decision rules to homogenize our clinical practices [16]. In addition, we can hypothesize that our study is premature in relation to the publication of these same guidelines in 2022. Not all emergency physicians have had time to get to know these new guidelines. Secondly, the median age of our cohort was 79 years (43–88). The risk stratification of traumatic intracranial hemorrhage is based on the collection of functional post-trauma signs (i.e.,: loss of consciousness, amnesia). However, nearly 30% of these older patients show cognitive impairment, making it difficult or impossible to collect these signs [17]. It is likely that physicians may indicate the realization of a head CT scan if there is any doubt.

**Table 1. Patients' characteristics admitted to the Emergency Department with ground-level fall-related minor head injury.**

|  | Overall N = 631 | Missing value |
|---|---|---|
| **Age, median (IQ1-IQ3)** | 79 (63-88) |  |
| **Sex, female** | 383 (60.7) |  |
| **Place of residence** |  | 5 (0.8) |
| Community-dwelling | 483 (76.6) |  |
| Nursing Homes | 137 (21.7) |  |
| Homelessness | 6 (1.0) |  |
| **Antiplatelets** | 154 (24.4) | 14 (2.2) |
| **Anticoagulants** | 141(22.4) | 15 (2.4) |
| **Post head injury symptoms** |  | 54 (8.6) |
| No symptom | 382 (60.5) |  |
| Amnesia | 39 (6.2) |  |
| Loss of consciousness | 69 (10.9) |  |
| Confusion | 72 (11.4) |  |
| Headache | 93 (14.7) |  |
| Seizure | 3 (0.5) |  |
| Vomiting | 37 (6.4) |  |
| **Clinical findings at the ED** |  |  |
| **Coma Glasgow Scale Score** |  |  |
| *15* | 587 (93.0) |  |
| *14* | 37 (5.9) |  |
| *13* | 7 (1.1) |  |
| **Visible head impact location** |  | 6 (0.9) |
| None | 155 (24.6) |  |
| Facial | 104 (16.5) |  |
| Frontal | 153 (24.3) |  |
| Temporal-parietal-occipital | 172 (27.3) |  |
| Multiple | 41 (6.5) |  |
| **Focal neurological signs** | 16 (2.5) | 16 (2.5) |
| **Pupillary abnormalities** | 6 (1.0) | 24 (3.8) |
| **Basal skull fracture signs** | 20 (3.2) | 16 (2.5) |
| **Fall precipitating factor** |  | 146 (23.1) |
| Environmental | 323 (51.2) |  |
| Syncope | 45 (7.1) |  |
| Faintness or vertigo | 81 (12.8) |  |
| Alcohol intoxication | 29 (4.6) |  |
| Others | 7 (1.1) |  |

This may be at the source of a variation in practices, explaining the wide variability in head CT scan prescription. Moreover, in our study, ED characteristics (type of institution, number of admissions per day, onsite neurosurgery unit) were not associated with a variation in the number of prescriptions. The literature reports divergent results on this aspect. For instance, regarding the use of head CT scan among patients with suspected mild TBI, Ryu et al. showed that a head CT scan was more frequently prescribed in urban EDs than in rural ones [18].

**Table 2. Patients' main characteristics according to French regions.**

| Provinces | Total | Type of institution, University Hospital (%) | Head CT scan use (%) | Age (IQ1-IQ3) | Sex, female (%) | Community-dwelling (%) | Antiplatelets (%) | Anticoagulants (%) | Coma Glasgow Scale score 15 (%) |
|---|---|---|---|---|---|---|---|---|---|
| Auvergne-Rhône Alpes | 92 | 45 (48.9) | 67.4 (CI95%: 56.8–76.8) | 78 (63.5-87.5) | 71.7 | 84.6 | 27.8 | 17.1 | 96.7 |
| Bourgogne-Franche-Comté | 13 | 6 (46.2) | 84.6 (CI95%: 54.5–98.1) | 87 (76-92) | 53.9 | 69.3 | 23.1 | 30.8 | 100 |
| Bretagne | 22 | 21 (95.5) | 72.7 (CI95%: 49.8–89.3) | 83 (69-85) | 68.2 | 81.8 | 28.6 | 33.3 | 95.4 |
| Centre – Val de Loire | 22 | 16 (72.7) | 45.5 (CI95%: 24.4–67.8) | 84.5 (64-91) | 40.9 | 81.8 | 27.3 | 9.1 | 95.4 |
| Grand Est | 62 | 40 (64.5) | 69.4 (CI95%: 58.7–66.7) | 78 (69-87) | 64.5 | 79 | 30.7 | 21.3 | 96.7 |
| Hauts-de-France | 33 | 0 | 60.6 (CI95%: 56.3–80.4) | 86 (71-93) | 75.8 | 69.7 | 17.8 | 27.6 | 96.9 |
| Normandie | 14 | 13 (92.9) | 57.1 (CI95%: 28.9–82.3) | 73 (66-86) | 71.4 | 92.9 | 21.4 | 35.7 | 92.9 |
| Nouvelle Aquitaine | 65 | 27 (41.5) | 72.3 (CI95%: 59.8–82.7) | 82.5 (72-87) | 49.2 | 79.7 | 24.6 | 30.8 | 86.2 |
| Occitanie | 106 | 59 (55.7) | 71.7 (CI95%: 62.1–82.0) | 77 (63-89) | 57.5 | 72.4 | 24.8 | 31.4 | 89.6 |
| Pays de la Loire | 26 | 16 (61.6) | 57.7 (CI95%: 36.9–76.6) | 83 (59-88) | 50 | 36 | 44 | 19.2 | 96.2 |
| Provence-Alpes-Côte-d'Azur | 46 | 10 (21.7) | 56.5 (CI95%: 58.7–66.7) | 88.5 (79-93) | 60.9 | 71.7 | 21.7 | 23.9 | 95.7 |
| Île-de-France | 101 | 63 (62.4) | 52.5 (CI95%: 42.3–62.5) | 76 (61-88) | 64.4 | 79 | 16 | 17.5 | 94.1 |
| Oversea regions | 29 | 29 (100) | 75.0 (CI95%: 56.4–89.7) | 68.4 (61-79) | 41.3 | 96.5 | 17.9 | 3.4 | 79.3 |

Several patients' characteristics were associated with head CT scan use. Antiplatelet and anticoagulant agents were the factors mostly associated with a head CT scan performance in our cohort. These results are probably explained by systematic indication, according to national guidelines, for head CT scan in anticoagulated patients [13]. In our cohort, almost half of the patients had preinjury antiplatelets or anticoagulants. These results are in line with those of the study performed by O'Brien et al. in which 70% of patients had preinjury antiplatelets or anticoagulants [19]. Recently, several studies have suggested that preinjury antiplatelets or anticoagulants medication may be not associated with traumatic ICH in patients after ground-level fall with head injury [20].

Furthermore, our study suggested that the rate of appropriate head CT scans in this population was low, and their use was not correlated with the traumatic ICH yield rate. These findings are in line with a study performed in 40 pediatric EDs in the US. Despite showing a large variability of head CT scan in minor head injured patients across pediatric EDs (ranged 19–58%), high rate of prescriptions was not associated with high rate of traumatic ICH [21].

Taken together, these results should incite us to use a clinical decision rule dedicated to this population. For instance, De Wit et al. derived the FALLS decision rule from a prospective multicentric study carried out in Canada and in the USA [22]. This rule has been recently prospectively validated [23]. This rule indicates that head CT scan may not be required if 1) patients did not hit their head when they fell, 2) no amnesia of the fall 3) no new abnormality on neurological examination, and 4) the Clinical Frailty Scale Score is < 5. Despite its low specificity (20.3%) the Falls Decision Rule might safely avoid head CT scans in 20% of patients. Interestingly, neither anti-platelets nor anticoagulants are mentioned in this rule.

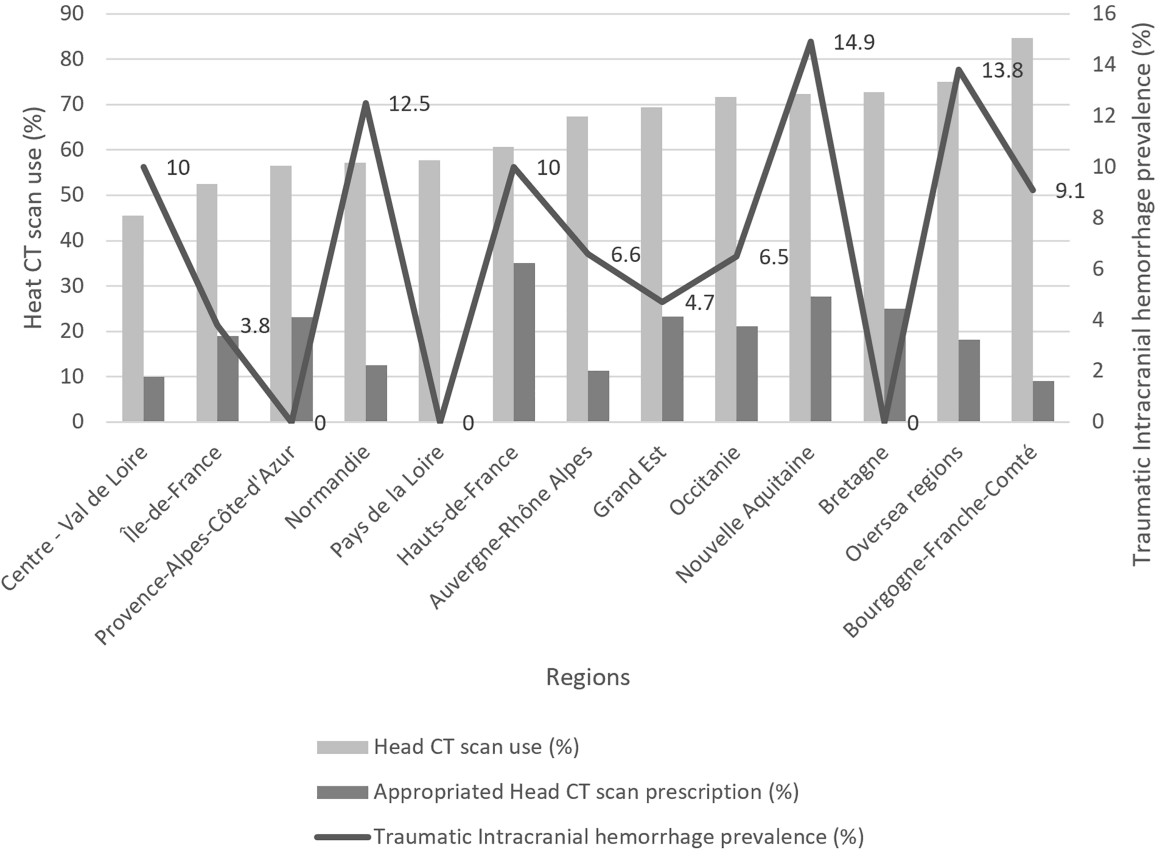

**Fig 2. Traumatic intracranial hemorrhage yield rate according to head CT scan use at the regional level in patients presenting to Emergency Departments for head injury related to ground-level fall.**

**Table 3. Head CT scan variation use in subpopulations at the regional level.**

| | Total | Head CT scan use prevalence | Region level Head CT scan use variation | ICH prevalence |
|---|---|---|---|---|
| Minor head injury, GCS[a] score 13–15 | 631 | 64.8 (CI95%: 60.1–68.6) | 45.5% (CI95%: 24.4–67.8) – 84.6% (CI95% 54.5–98.1) | 7.10% |
| Minor head injury, GCS 15, without focal neurological signs | 575 | 62.8 (CI95%: 58.7–66.7) | 47.6% (CI95%: 25.7–70.2) – 84.6% (CI95% 54.5–98.1) | 5% |
| Minor head injury, GCS 15, neither focal neurological sign nor anticoagulants | 450 | 54.0 (CI95%: 49.3–58.7) | 25.0% (CI95%: 3.2–65.1) – 77.8% (CI95% 40.0–97.2) | 5.70% |
| Minor head injury, GCS 15, neither focal neurological signs, anticoagulants nor antiplatelets | 323 | 40.6 (CI95%: 35.2–46.1) | 11.1% (CI95%: 2.0–48.0) – 66.6% (CI95% 22.0–95.0) | 4.60% |

[a]GCS: Glasgow Coma Scale score; ICH: intracranial hemorrhage.

The main strength of our study resides in its prospective nature, enabling an accurate description of clinical practices in usual care settings. However, in the absence of patients' follow-up after 48 hours, the assessment of head CT scan may be questionable, as some patients may have had a head CT scan after the follow-up period. Nevertheless, this is limited by the fact that national guidelines recommend a CT scan prescription within 8 hours after the trauma [13]. Moreover, the

**Table 4. Predictive factors associated with head CT scan use at the Emergency Department in patients with ground-level fall-related minor head injury.**

|  | Odds ratio | CI 95% | p-value |
|---|---|---|---|
| Age, per year | 1.0 | 0.9-1.1 | 0.26 |
| Antiplatelets | 29.2 | 12.2 - 69.9 | <0.001 |
| Anticoagulants | 69.9 | 20.0 - 243.9 | <0.001 |
| <u>Fall precipitating factor</u> |  |  |  |
| Syncope | 6.9 | 2.0 - 24.2 | <0.01 |
| Faintness or vertigo | 0.9 | 0.4 - 2.0 | 0.81 |
| Alcohol intoxication | 3.4 | 0.90 - 12.8 | 0.07 |
| Others | 0.8 | 0.1 - 8.8 | 0.84 |
| <u>Clinical findings at the ED</u> |  |  |  |
| Amnesia | 3.2 | 1.0 - 10.5 | 0.05 |
| Loss of consciousness | 5.6 | 2.0 - 15.9 | <0.01 |
| Confusion | 2.1 | 0.7 - 6.1 | 0.17 |

prevalence of traumatic ICH may have been underestimated, since not all patients received a CT scan. Furthermore, the small sample sizes reported for certain regions (e.g., Bourgogne, Normandie) may limit the power of our statistical analyses.

## Conclusions

Head CT scan use in patient presenting to EDs with head injuries after ground-level falls is highly variable. High rate of head CT scan use is not correlated with high traumatic intracranial hemorrhage yield rate. The prescription of appropriate head CT scan is low in this population. These results support the use of a clinical decision rule dedicated to this population in order to standardize our practices.

## Supporting information

**S1 Table. Head CT scan use following Emergency Departments' characteristics.**
(DOCX)

**S2 Table. Head CT scan use following patients' characteristics.**
(DOCX)

**S3 Table. Predictive factors associated with head CT scan use at the Emergency Department in patients with ground-level fall-related minor head trauma presenting with Glasgow Coma scale score 15 neither focal neurologic sign nor anticoagulant using mixed logistic regression.**
(DOCX)

**S4 Table. Predictive factors associated with head CT scan use at the Emergency Department in patients with ground-level fall-related minor head trauma presenting with Glasgow Coma scale score 15 without focal neurologic sign using mixed logistic regression.**
(DOCX)

**S5 Table. Predictive factors associated with head CT scan use at the Emergency Department in patients with ground-level fall-related minor head trauma presenting with Glasgow Coma scale score 15 neither focal neurologic sign nor anticoagulant using mixed logistic regression.**
(DOCX)

## Acknowledgments

Thank you very much to Manon Hebrard, Clinical Research Coordinator at the University Hospital of Toulouse for her involvement in this study research and her valuable contribution to it. Thank you also to Isabelle Olivier, Regulatory Project Manager at Toulouse University Hospital, for her support with the administrative phase of the study. Thanks to all the hospital practitioners for helping us to include patients. Many thanks to each local investigating team of the EPI-TC IRU-SFMU network for their contribution in preparing and conducting this study. Thank you very much as well to Ana Inés Darquier, native medical writer, for proofreading this article in English.

IRU EPI-TC group (lead author: Thomas Lafon, thomas.lafon@chu-limoges.fr)

Lenglet Hugo[1], Schmit Hugo[2], Claveries Paul[3], Pereira Xavier[3], Claessens Yann-Erick[4], Desclefs Jean-Philippe[5], Delta Delphine[6], Lombart Aline[7], Schmit Hugo[8], Cohen Rudy[9], Laborne François-Xavier[10], Gouetta Alicia[11], Mourier Charlène[12], Duchenne Jonathan[12], Ayoub Touihar[13], Boulanger Bertrand[14], Broche Claire[14], Tran Duc-Minh[15], Sellami Maryam[16], Rym Hamed[17], Ben Hallouda Kassara[18], Emilie Gelin[19], Fabre Jean[20], Rosenblatt Julie[21], Gangneron Aude-Marie[22], Chomarat Chloé[23], Ombeline Susong[24], Ohayon Lisa[25], Gaouiaoui Rachid[26], Dessena Anastasia[26], Menard Bastien[26], Picaud Adrien[27], Bangala Gustave[28], Bosc Juliane[29], Romain Blondet[30], Darraillans Didier[30], Grgek Sylvie[31], Murgue Davy[31], Thomas Fabien[31], Jauriac Violetta[32], Chareyron Anne[33], Boulanger Alice[34], Gonfrere Gwendoline[34], Thiriez Sylvain[35], Hurgon Alice[36], Flambard Maud[37], Senon-LalogeAnna[38], Bauduin Tristan[38], Berton Laurence[39], Clotilde Cazenave[39], Maiello Ernesto[40], Bac Arnaud[41], Bizouard Thomas[42], Hicheri Yassine[43], Daniel Matthieu[44], Faivre Caleb[45], Clement Bénédicte[46], Karam Henri Hani[47], Giacomin Gaëtan[48], Coisy Fabien[49], Guenezan Jeremy[50], Sugranes Pauline[51], Roussel Mélanie[52], Le Borgne Pier-rick[53], Laribi Said[54], Abou-Badra Mathieu[55], Aymeric Bodineau[56], Charney Alexandre[57], Nguyen Van Tinh Meryl[58], Laporte Hadrien[59], Yahoui Yahia[59], Jacquin Laurent[60], Chocron Richard[61], Le Bail Gaëlle[62], Sende Jean[63], Olivier Thomas[64], Levy Delphine[65], Gerlier Camille[66], Addou Sarah[67], Gautier Maxime[68], Burggraff Eric[69], Cancella De Abreu Marta[70], Aubertein Pierre[71], Thomas Lafon[72], Romain Adam[73], Solene Loth[73], Flore Tabaka[74], Florian Negrello[75], Mustapha Sebbane[76]

[1] Bataillon de marins-pompiers de Marseille, Marseille, France

[2] Centre hospitalier Annecy Genevois, Saint Julien en Genevois, France

[3] Centre hospitalier de la Côte Basque, Bayonne, France

[4] Centre hospitalier Princesse Grace, Monaco, France

[5] Centre hospitalier Sud Francilien, Corbeil-Essonnes, France

[6] Centre hospitalier Universitaire de la Guadeloupe, Pointe-à-Pitre, France

[7] Centre hospitalier Albi, Albi, France

[8] Centre hospitalier Annecy Genevois, Annecy, France

[9] Centre hospitalier Argenteuil, Argenteuil, France

[10] Centre hospitalier Arpajon, Arpajon, France

[11] Centre hospitalier Auch, Auch, France

[12] Centre hospitalier Aurillac, Aurillac, France

[13] Centre hospitalier Auxerre, Auxerre, France

[14] Centre hospitalier Bretagne Atlantique, Vannes, France

[15] Centre hospitalier Cahors, Cahors, France

[16] Centre hospitalier Charleville-Mezieres, Charleville-Mezieres, France

[17] Centre hospitalier Chartres, Chartres, France

[18] Centre hospitalier Colmar, Colmar, France

[19] Centre hospitalier Cornouaille, Quimper, France

[20] Centre hospitalier Dax, Dax, France

[21] Centre hospitalier de Bigorre, Tarbes, France

[22] Centre hospitalier de Bourg-en-Bresse, Bourg-en-Bresse, France

23 Centre hospitalier De Gonesse, Gonesse, France

24 Centre hospitalier de Rochefort, Rochefort, France

25 Centre hospitalier de Verdun Saint-Mihiel, Verdun, France

26 Centre hospitalier Dreux, Dreux, France

27 Centre hospitalier Le Mans, Le Mans, France

28 Centre hospitalier Mantes-la-Jolie, Mantes-la-Jolie, France

29 Centre hospitalier Marmande, Marmande, France

30 Centre hospitalier Mont-de-Marsan, Mont-de-Marsan, France

31 Centre hospitalier Moulins, Moulins, France

32 Centre hospitalier Nogent-le-Rotrou, Nogent-le-Rotrou, France

33 Centre hospitalier Pontoise, Pontoise, France

34 Centre hospitalier Rodez, Rodez, France

35 Centre hospitalier Roubaix, Roubais, France

36 Centre hospitalier Saint-Gaudens, Saint-Gaudens, France

37 Centre hospitalier Sud Seine et Marne, Fontainebleau, France

38 Centre hospitalier Toulon, Toulon, France

39 Centre hospitalier Versailles, Versailles, France

40 Centre hospitalier Vienne, Vienne, France

41 Centre hospitalier Lillebonne, Lillebonne, France

42 CHU Angers, Angers, France

43 CHU Clermont Ferrand, Clermont Ferrand, France

44 CHU de La Réunion Site Nord Félix Guyon, Saint-Denis, France

45 CHU Dijon, Dijon, France

46 CHU Hospice de Lyon Croix Rousse, Lyon, France

47 CHU Limoges, Limoges, France

48 CHU Nancy, Nancy, France

49 CHU Nîmes, Nîmes, France

50 CHU Poitiers, Poitiers, France

51 CHU Rennes, Rennes, France

52 CHU Rouen, Rouen, France

53 CHU Strasbourg, Strasbourg, France

54 CHU Tours, Tours, France

55 Clinique des Cèdres, Cornebarrieu, France

56 Clinique la Croix du Sud, Quint-Fonsegrives, France

57 Grand Hôpital de l'Est Francilien, Meaux, France

58 Hôpital Avicenne, Bobigny, France

59 Hôpital Beaumont sur Oise, Beaumont sur Oise, France

60 Hôpital Edouard Herriot – Hospices Civils de Lyon, Lyon, France

61 Hôpital européen Georges-Pompidou, Paris, France

62 Hôpital Garches Raymond Poincaré, Garches, France

63 Hôpital privé Armand Brillard, Nogent-sur-Marne, France

64 Hôpital Saint Antoine, Paris, France

65 Hôpital Saint-Joseph, Marseille, France

66 Hôpital Saint-Joseph Paris, Paris, France

67 Hôpital Saint-Louis, Paris, France

68 Hôpital Simone Veil, Eaubonne, France

69 Hôpital Tenon, Paris, France

70 Hôpital Universitaire Pitié-Salpêtrière, Paris, France

71 Médipôle Lyon-Villeurbanne, Villeurbanne, France

72 CHU Limoges, Limoges France

73 CHU Besançon, Besançon, France

74 CH Beauvais, Beauvais, France

75 CHU Martinique, Fort-de-France, France

76 CHU Montpellier, Montpellier, France

## Author contributions

**Conceptualization:** Xavier Dubucs, Frederic Balen, Sandrine Charpentier, Marcel Émond.

**Formal analysis:** Xavier Dubucs, Frederic Balen, Pierre-Hugues Cormicael, Sandrine Charpentier.

**Funding acquisition:** Xavier Dubucs.

**Investigation:** Xavier Dubucs.

**Methodology:** Xavier Dubucs, Frederic Balen, Pierre-Hugues Cormicael, Axel Benhamed, Éric Mercier, Sandrine Charpentier, Marcel Émond.

**Project administration:** Frederic Balen.

**Supervision:** Sandrine Charpentier, Marcel Émond.

**Validation:** Pierre-Hugues Cormicael, Valérie Boucher, Éric Mercier.

**Writing – original draft:** Xavier Dubucs.

**Writing – review & editing:** Xavier Dubucs, Frederic Balen, Pierre-Hugues Cormicael, Axel Benhamed, Valérie Boucher, Éric Mercier, Sandrine Charpentier, Marcel Émond.

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
