## [Decision Letter · Decision Letter 0]

7 Aug 2025

Dear Dr. Dubucs,

Thank you for submitting your manuscript to PLOS ONE. After careful consideration, we feel that it has merit but does not fully meet PLOS ONE’s publication criteria as it currently stands. Therefore, we invite you to submit a revised version of the manuscript that addresses the points raised during the review process.

We look forward to receiving your revised manuscript.

Kind regards,

Dr. Jan Chrusciel

Academic Editor

PLOS ONE

Journal Requirements:

2. In the ethics statement in the Methods, you have specified that verbal consent was obtained. Please provide additional details regarding how this consent was documented and witnessed, and state whether this was approved by the IRB.

3. Please include a complete copy of PLOS’ questionnaire on inclusivity in global research in your revised manuscript. Our policy for research in this area aims to improve transparency in the reporting of research performed outside of researchers’ own country or community. The policy applies to researchers who have travelled to a different country to conduct research, research with Indigenous populations or their lands, and research on cultural artefacts. The questionnaire can also be requested at the journal’s discretion for any other submissions, even if these conditions are not met. Please find more information on the policy and a link to download a blank copy of the questionnaire here: https://journals.plos.org/plosone/s/best-practices-in-research-reporting. Please upload a completed version of your questionnaire as Supporting Information when you resubmit your manuscript.

5. Thank you for stating in your Funding Statement:

[This study was supported by the French Emergency Medicine Research Network (Initiatives de Recherche aux Urgences [IRU group]) of the Société Française de Médecine d’Urgence (SFMU). The funder had no role in the design, data collection, data analysis, and reporting of this study.].

6. Thank you for stating the following in your manuscript:

[This study was supported by the French Emergency Medicine Research Network (Initiatives de Recherche aux Urgences [IRU group]) of the Société Française de Médecine d’Urgence (SFMU). The funder had no role in the design, data collection, data analysis, and reporting of this study.]

[This study was supported by the French Emergency Medicine Research Network (Initiatives de Recherche aux Urgences [IRU group]) of the Société Française de Médecine d’Urgence (SFMU). The funder had no role in the design, data collection, data analysis, and reporting of this study.]

7. In the online submission form, you indicated that [The data on which the results presented in the study are based are available on request from Xavier Dubucs.].

8. When completing the data availability statement of the submission form, you indicated that you will make your data available on acceptance. We strongly recommend all authors decide on a data sharing plan before acceptance, as the process can be lengthy and hold up publication timelines. Please note that, though access restrictions are acceptable now, your entire data will need to be made freely accessible if your manuscript is accepted for publication. This policy applies to all data except where public deposition would breach compliance with the protocol approved by your research ethics board. If you are unable to adhere to our open data policy, please kindly revise your statement to explain your reasoning and we will seek the editor's input on an exemption. Please be assured that, once you have provided your new statement, the assessment of your exemption will not hold up the peer review process.

9. One of the noted authors is a group or consortium [IRU EPI-TC group]. In addition to naming the author group, please list the individual authors and affiliations within this group in the acknowledgments section of your manuscript. Please also indicate clearly a lead author for this group along with a contact email address.

10. We note that Figures 2 and S1 in your submission contain map images which may be copyrighted. All PLOS content is published under the Creative Commons Attribution License (CC BY 4.0), which means that the manuscript, images, and Supporting Information files will be freely available online, and any third party is permitted to access, download, copy, distribute, and use these materials in any way, even commercially, with proper attribution. For these reasons, we cannot publish previously copyrighted maps or satellite images created using proprietary data, such as Google software (Google Maps, Street View, and Earth). For more information, see our copyright guidelines: http://journals.plos.org/plosone/s/licenses-and-copyright.

1. You may seek permission from the original copyright holder of Figures 2 and S1 to publish the content specifically under the CC BY 4.0 license. 

Please upload the completed Content Permission Form or other proof of granted permissions as an "Other" file with your submission

11. We notice that your supplementary figure is uploaded with the file type 'Figure'. Please amend the file type to 'Supporting Information'. Please ensure that each Supporting Information file has a legend listed in the manuscript after the references list.

Reviewers' comments:

Reviewer's Responses to Questions

**Comments to the Author**

1. Is the manuscript technically sound, and do the data support the conclusions?

Reviewer #1: Partly

Reviewer #2: Partly

2. Has the statistical analysis been performed appropriately and rigorously?

Reviewer #1: I Don't Know

Reviewer #2: Yes

3. Have the authors made all data underlying the findings in their manuscript fully available?

Reviewer #1: Yes

Reviewer #2: Yes

4. Is the manuscript presented in an intelligible fashion and written in standard English?

Reviewer #1: Yes

Reviewer #2: No

Reviewer #1: Thank you for the opportunity to review this manuscript. The proposed topic highlights several important issues concerning a subject with significant consequences for patients. I hope these comments will help the authors.

Title:

The title does not specify that this is a secondary analysis of an already published article.

Abstract:

L 136: The described outcomes does not correspond to the study's outcomes.

Introduction:

L 156: Reference 1 could be supplemented with a more recent article by Andrew I R Maas et al, "Traumatic brain injury: progress and challenges in prevention, clinical care, and research," The Lancet Neurology, 2022.

L 161: References 3 and 4 should be positioned in the previous sentence (L159).

L 169: Reference should be supplemented by Majdan et al, "Epidemiology of traumatic brain injuries in Europe: a cross-sectional analysis," The Lancet Public Health, 2016.

L 170: The publication by Mori et al, "Indications for Computed Tomography in Older Adult Patients With Minor Head Injury in the Emergency Department," Academic Emergency Medicine, 2020, is a more recent reference.

Materials and Methods:

L 182: 71 EDs participated in the original study ("Head injuries in prehospital and Emergency Department settings: a prospective multicentre cross sectional study in France" by Dubucs et al., BMC Emergency Medicine, 2024), but only 63 are included in this study. What explains the difference? Did some centers not perform CT scans?

L 187 to 191: It would be preferable to cite those that are missing. It would be interesting to have a table by ED “Type of institution” meaning a mix of tables S1 and S2.

L 211: What data were collected regarding the patient's hospital stay?

Results:

L 251: In the original study, there were 636 patients who had a CT scan, but in this article, there are 631 patients included.

L 253: According to Table 1, the number of patients without post-TBI symptoms does not correspond.

L 256: The results of the study's second objective are not sufficiently highlighted. They only appear in Figure 3. The proportions of patients who received appropriate CT scan prescriptions according to traumatic ICH risk (high and intermediate) are missing, which is the secondary objective.

L 259: What is the number of patients with unknown mechanism? This does not appear in the flow chart.

L 261: Table 1: What do the missing values for focal neurological signs, pupillary abnormalities, and basal skull fracture signs represent? Table S3 provides the same information as Table 1 but presented more relevantly; it would be preferable to exchange them.

L 264: Table S1: What are the arguments for stating that at the regional level, patient characteristics are similar? In this same table, would it be possible to add a column with the "Type of institution" variable? Furthermore, it is difficult to establish a link between heterogeneity in CT scan prescriptions at the regional level and heterogeneity in the practices of clinicians working in emergency departments

L 266: Figure 2 - the p-value does not appear on the map. The regional level lacks granularity. The type of institution by region should be displayed.

L 273: Table 2 – legend is missing

L 278: How is this correlation determined?

L 281: No p-value on the graph in Figure 3. The regional level does not facilitate reading of results because some regions included few patients (Bourgogne FC, Normandie) and strongly impact the results shown in Figure 3. Furthermore, is the proportion of appropriate head CT scan prescriptions based on the total patients admitted to the ED after ground-level fall-related minor head injury or on the proportion of head CT scan use?

Discussion :

L 296: Does the heterogeneity concern CT use among patients admitted to the ED after ground-level fall-related minor head injury? It appears that even for appropriate prescriptions, there is no correlation with ICH incidence.

L 298: "Our study also suggests that this variation may be explained by patients' characteristics." I do not understand this suggestion - CT scan prescription recommendations are based on patient characteristics.

L 301: I completely agree with you that the main problem concerns implementation science, as explained by Cabana MD et al. "Why don't physicians follow clinical practice guidelines? A framework for improvement." JAMA. 1999, or Tucker et al. "Implementation Science: Application of Evidence-Based Practice Models to Improve Healthcare Quality." Worldviews Evid Based Nurs. 2021. The regional level does not seem relevant to explain heterogeneity. It is probably more at the individual level and how each clinician balances their clinical experience with recommendations issued by scientific societies, which are two of the pillars of EBM.

L 315: A high rate of CT prescription is not associated with ICH prevalence, but is a high rate of appropriate prescriptions associated with a high rate of ICH?

L 325: Where does this appear in your results?

L 349: This statement is particularly true for elderly patients who are not prioritized for examinations like CT scans and who may have them at a distance from the trauma. This data is particularly important to highlight given the mean age of the population in this study.

Reviewer #2: This is a multicenter cross-sectional study done in France to evaluate the variations of CTH use in patients who sustained a GLF-related head injury. The secondary objective was to measure the rate of appropriate CTH orders and to identify characteristics associated with CTH orders

Major comments

1. Need clarification on this sentence in the beginning of results: "A head CT scan was performed in 409 patients (64.8%, CI95% 61.0-68.5) and a traumatic ICH was reported in 29 (4.6%, CI95% 3.1-6.5) of them." is 4.6% out of 631 patients in the study or out of 409 patients who received CT? I am reading it as ICH yield on all CTs ordered so it should be 7.1% (29/409). In addition, Table 2 is reporting ICH prevalence out of the total of 631 patients (cohort) as 7.1%, which is confusing here. Need to reconcile those percentages

2. In discussion, author concluded that : "This heterogeneity was not associated with significant

difference of traumatic ICH prevalence." This is a bit confusing. traumatic ICH is there whether a CTH is ordered or not. In other words, for patients in the cohort who did not received a CT, they might have a traumatic ICH even if they didn't get the CTH, and would still contribute to the prevalence technically but it's just not known. Should consider changing that sentence to "traumatic ICH yield." Same apply to Conclusion and implications and the rest of the paper.

3. Authors use "appropriate CTH orders" numerously throughout the paper. It is assumed that it's any CTH ordered that concordant with National Guidelines (Study design and Settings). It might be good to define what is considered appropriate or inappropriate for this study since it's not explicitly stated.

Minor comments

1. Recommend going back to review spelling and grammar for the whole paper: Line 175 "The secondary objective was to measure the

prescription rate of appropriated head CT scan and to identify patients’ and EDs characteristic"

2. Coma Glasgow Scale Score in Table 1 should say Glasgow Coma Scale Score

**Do you want your identity to be public for this peer review?** For information about this choice, including consent withdrawal, please see our Privacy Policy

Reviewer #1: No

Reviewer #2: No

---

## [Author Response · Author response to Decision Letter 1]

22 Sep 2025

September 12, 2025

To Dr. Jan Chrusciel, Academic Editor,

Plos One

Ref: PONE-D-25-25181e

Title: Variation in the use of head CT scan in patients attending the emergency department after ground-level fall-related minor head injury

Dear Editor,

Thank you for o�ering us the opportunity to revise our manuscript “Variation in the use of head CT scan in patients attending the emergency department after ground-level fall-related minor head injury” and resubmit it for publication in “Plos One”. We have revised the paper according to the Editorial and Reviewer’s comments. In this letter, we address the changes made point by point.

We would like to thank the Reviewers for his/her comments, which have given us the opportunity to further improve our manuscript. We trust that you will find our revisions acceptable and look forward to hearing from you.

Yours sincerely,

Xavier DUBUCS

Editorial comments.

Journal Requirements:

Response:

We have made all necessary changes in accordance with these recommendations.

2. In the ethics statement in the Methods, you have specified that verbal consent was obtained. Please provide additional details regarding how this consent was documented and witnessed, and state whether this was approved by the IRB.

Response:

We have enhanced the clarity of the Ethics Statement section:

“According to French and European law on ethics, all included patients were informed that their anonymized data will be used for the study. The oral non-opposition of all patients was collected and a written information notice was given to each included patient explaining the objectives of the study and the legislation explaining this research. This non-opposition was documented in the patient's medical file. According to the French ethics and regulatory law (public health code) prospective studies based on the exploitation of usual care data don’t should be submit at an ethics committee (IRB) but they have to be declared or covered by reference methodology of the French National Commission for Informatics and Liberties (CNIL). Toulouse University Hospital signed a commitment of compliance to the reference methodology MR-004 of the French National Commission for Informatics and Liberties (CNIL). After evaluation and validation by the data protection officer and according to the General Data Protection Regulation (Regulation EU 2016/679 of the European Parliament and of the Council of 27 April 2016), this study completing all the criteria, it is register in the register of data study of the Toulouse University Hospital (number’s register: RnIPH 2022–78) and cover by the MR-004 (CNIL number: 2206723 v 0). This study was approved by Toulouse University Hospital and confirms that ethical requirements were totally respected in the above report.”

3. Please include a complete copy of PLOS’ questionnaire on inclusivity in global research in your revised manuscript. Our policy for research in this area aims to improve transparency in the reporting of research performed outside of researchers’ own country or community. The policy applies to researchers who have travelled to a different country to conduct research, research with Indigenous populations or their lands, and research on cultural artefacts. The questionnaire can also be requested at the journal’s discretion for any other submissions, even if these conditions are not met. Please find more information on the policy and a link to download a blank copy of the questionnaire here: https://journals.plos.org/plosone/s/best-practices-in-research-reporting. Please upload a completed version of your questionnaire as Supporting Information when you resubmit your manuscript.

Response:

This document has been completed.

Response:

This grant number has been added

5. Thank you for stating in your Funding Statement:

[This study was supported by the French Emergency Medicine Research Network (Initiatives de Recherche aux Urgences [IRU group]) of the Société Française de Médecine d’Urgence (SFMU). The funder had no role in the design, data collection, data analysis, and reporting of this study.].

Response:

These sentences have been added.

6. Thank you for stating the following in your manuscript:

[This study was supported by the French Emergency Medicine Research Network (Initiatives de Recherche aux Urgences [IRU group]) of the Société Française de Médecine d’Urgence (SFMU). The funder had no role in the design, data collection, data analysis, and reporting of this study.]

[This study was supported by the French Emergency Medicine Research Network (Initiatives de Recherche aux Urgences [IRU group]) of the Société Française de Médecine d’Urgence (SFMU). The funder had no role in the design, data collection, data analysis, and reporting of this study.]

Response:

This chapter has been removed from the manuscript and the cover letter has been modified to reflect the previous suggestions.

7. In the online submission form, you indicated that [The data on which the results presented in the study are based are available on request from Xavier Dubucs.].

Response:

We have changed our answers to these questions.

8. When completing the data availability statement of the submission form, you indicated that you will make your data available on acceptance. We strongly recommend all authors decide on a data sharing plan before acceptance, as the process can be lengthy and hold up publication timelines. Please note that, though access restrictions are acceptable now, your entire data will need to be made freely accessible if your manuscript is accepted for publication. This policy applies to all data except where public deposition would breach compliance with the protocol approved by your research ethics board. If you are unable to adhere to our open data policy, please kindly revise your statement to explain your reasoning and we will seek the editor's input on an exemption. Please be assured that, once you have provided your new statement, the assessment of your exemption will not hold up the peer review process.

Response:

I agree, if my article is accepted, to make my data freely available.

9. One of the noted authors is a group or consortium [IRU EPI-TC group]. In addition to naming the author group, please list the individual authors and affiliations within this group in the acknowledgments section of your manuscript. Please also indicate clearly a lead author for this group along with a contact email address.

Response:

Done

10. We note that Figures 2 and S1 in your submission contain map images which may be copyrighted. All PLOS content is published under the Creative Commons Attribution License (CC BY 4.0), which means that the manuscript, images, and Supporting Information files will be freely available online, and any third party is permitted to access, download, copy, distribute, and use these materials in any way, even commercially, with proper attribution. For these reasons, we cannot publish previously copyrighted maps or satellite images created using proprietary data, such as Google software (Google Maps, Street View, and Earth). For more information, see our copyright guidelines: http://journals.plos.org/plosone/s/licenses-and-copyright.

Response:

We have decided to remove these two figures because we are unable to provide copyright information.

1. You may seek permission from the original copyright holder of Figures 2 and S1 to publish the content specifically under the CC BY 4.0 license.

Please upload the completed Content Permission Form or other proof of granted permissions as an "Other" file with your submission

11. We notice that your supplementary figure is uploaded with the file type 'Figure'. Please amend the file type to 'Supporting Information'. Please ensure that each Supporting Information file has a legend listed in the manuscript after the references list.

Response:

These changes have been made.

Reviewers' comments:

Reviewer's Responses to Questions

Comments to the Author

1. Is the manuscript technically sound, and do the data support the conclusions?

Reviewer #1: Partly

Reviewer #2: Partly

2. Has the statistical analysis been performed appropriately and rigorously?

Reviewer #1: I Don't Know

Reviewer #2: Yes

3. Have the authors made all data underlying the findings in their manuscript fully available?

Reviewer #1: Yes

Reviewer #2: Yes

4. Is the manuscript presented in an intelligible fashion and written in standard English?

Reviewer #1: Yes

Reviewer #2: No

5. Review Comments to the Author

Reviewer #1:

Thank you for the opportunity to review this manuscript. The proposed topic highlights several important issues concerning a subject with significant consequences for patients. I hope these comments will help the authors.

Title:

The title does not specify that this is a secondary analysis of an already published article.

Response:

Please find below a new proposal for the title:

Variability in head computed tomography use for minor head injury after ground-level falls in the emergency department: a subanalysis of EPI-TC study

Abstract:

L 136: The described outcomes does not correspond to the study's outcomes.

Response:

Indeed, there was some inaccuracy, we have harmonized this in the abstract, the introduction, and the Methods section.

« The aim of this study was to assess the variation in the use of head CT scan in patients attending EDs with ground-level fall-related minor head injury. Secondary objectives were: i) to measure the prescription rate of appropriated head CT scan, ii) to identify patients’ and EDs characteristics associated with head CT scan prescription iii) to explore potential correlation between head CT scan use and traumatic intracranial hemorrhage (ICH) yield rate in this population. »

Introduction:

L 156: Reference 1 could be supplemented with a more recent article by An

---

## [Editor Report · Decision Letter 1]

29 Sep 2025

Variability in head computed tomography use for minor head injury after ground-level falls in the emergency department: a subanalysis of EPI-TC study

PONE-D-25-25181R1

Dear Dr. Dubucs,

We’re pleased to inform you that your manuscript has been judged scientifically suitable for publication and will be formally accepted for publication once it meets all outstanding technical requirements.

Kind regards,

Dr. Jan Chrusciel, M.D.

Academic Editor

PLOS ONE

Additional Editor Comments:

During the final stages of proofreading, please take care to distinguish between the words "appropriated" and "appropriate". A CT-scan prescription can be "appropriate", but not "appropriated".

---

## [Editor Report · Acceptance letter]

PONE-D-25-25181R1

PLOS One

Dear Dr. Dubucs,

I'm pleased to inform you that your manuscript has been deemed suitable for publication in PLOS One. Congratulations! Your manuscript is now being handed over to our production team.

Kind regards,

on behalf of

Dr. Jan Chrusciel

Academic Editor

PLOS One